Digital mapping of soil texture in ecoforest polygons in Quebec, Canada

Duchesne Louis louis.duchesne@mffp.gouv.qc.ca
Ouimet Rock
Direction de la recherche forestière, Ministère des Forêts, de la Faune et des Parcs , Québec , Québec , Canada
Fernandez-Marcos Maria Luisa
Electronic publication date: 2021 Jun 23
Publication date: 2021
Volume: 9
Electronic Location ID: e11685
Received 2021 Feb 25; Accepted 2021 Jun 7
Copyright: ©2021 Duchesne and Ouimet
Copyright year: 2021
Copyright holder: Duchesne and Ouimet
License: This is an open access article distributed under the terms of the Creative Commons Attribution License, which permits unrestricted use, distribution, reproduction and adaptation in any medium and for any purpose provided that it is properly attributed. For attribution, the original author(s), title, publication source (PeerJ) and either DOI or URL of the article must be cited.
License URL: https://creativecommons.org/licenses/by/4.0/

Keywords: Machine learning, Soil mapping, Soil texture, Random forest, Photo interpretation, Spatial data, Geostatistics, Soil samples, Forest inventory, Soil particle size

Funding: The Ministère des Forêts, de la Faune et des Parcs du Québec This research was supported by the Ministère des Forêts, de la Faune et des Parcs du Québec. The funders had no role in study design, data collection and analysis, decision to publish, or preparation of the manuscript.

==============================
Texture strongly influences the soil’s fundamental functions in forest ecosystems. In response to the growing demand for information on soil properties for environmental modeling, more and more studies have been conducted over the past decade to assess the spatial variability of soil properties on a regional to global scale. These investigations rely on the acquisition and compilation of numerous soil field records and on the development of statistical methods and technology. Here, we used random forest machine learning algorithms to model and map particle size composition in ecoforest polygons for the entire area of managed forests in the province of Quebec, Canada. We compiled archived laboratory analyses of 29,570 mineral soil samples (17,901 sites) and a set of 33 covariates, including 22 variables related to climate, five related to soil characteristics, three to spatial position or spatial context, two to relief and topography, and one to vegetation. After five repeats of 5-fold cross-validation, results show that models that include two functionally independent values regarding particle size composition explain 60%, 34%, and 78% of the variance in sand, silt and clay fractions, respectively, with mean absolute errors ranging from 4.0% for the clay fraction to 9.5% for the sand fraction. The most important model variables are those observed in the field and those interpreted from aerial photography regarding soil characteristics, followed by those regarding elevation and climate. Our results compare favorably with those of previous soil texture mapping studies for the same territory, in which particle size composition was modeled mainly from rasterized climatic and topographic covariates. The map we provide should meet the needs of provincial forest managers, as it is compatible with the ecoforest map that constitutes the basis of information for forest management in Quebec, Canada.

Introduction

Texture strongly influences the soil’s fundamental functions in forest ecosystems. In particular, the relative content of particles within specific size ranges affects soil mineral weathering rates (Kolka, Grigal & Nater, 1996), ion exchange and buffering capacity (Wiklander, 1975), as well as sequestration of nitrogen and carbon (Silver et al., 2000; Telles et al., 2003; Callesen et al., 2007). All these processes influence nutrient pools and cycling in forest ecosystems. Soil texture also affects soil water holding capacity, water uptake by plants and the overall hydrological cycle (Kern, 1995; Hultine et al., 2006; Saxton & Rawls, 2006), as well as the physical stability and supporting functions of soils (Coutts, 1983; Ruel, 1995; Schenk & Jackson, 2005). The importance of soil texture for the functioning of terrestrial and forest ecosystems is highlighted by the fact that it is a key part of various types of models that describe tree growth (Gustafson et al., 2017; D’Orangeville et al., 2018), species distribution (Williams et al., 1996; Itoh et al., 2003), forest disturbances (Schulte et al., 2005; Pourghasemi, 2016), biogeochemistry (Sverdrup & Warfvinge, 1993; Webb, Rosenzweig & Levine, 1993), hydrology (Yin & Arp, 1993), and land surface (Verseghy, 2007), among others.

In response to the growing demand for information on soil properties for environmental modeling, more and more studies have been conducted over the past decade to assess the spatial variability of soil properties on a regional to global scale (e.g., Grimm et al., 2008; Hong et al., 2013; Liao et al., 2013; Mansuy et al., 2014; Forkuor et al., 2017; Hengl et al., 2017). These investigations rely on the acquisition and compilation of numerous soil field records and on the development of statistical methods and technology that allow users to compute consistent and reliable spatial predictions of soil properties at various spatial scales (McBratney, Mendonça Santos & Minasny, 2003; Sanchez et al., 2009; Arrouays et al., 2014; Minasny & McBratney, 2016; Malone, Minasny & McBratney, 2017; Hengl & MacMillan, 2019).

The most advanced soil mapping methods involve producing predictions using optimal statistical models that define statistical relationships between observed soil properties and a set of rasterized environmental covariates that are relevant to explain the distribution of soil properties in the entire area to be mapped (Malone, Minasny & McBratney, 2017; Hengl & MacMillan, 2019). These covariates generally include information related to climate, vegetation, relief and topography, parent material, geological age, spatial or geographic position, and human or anthropogenic influences (McBratney, Mendonça Santos & Minasny, 2003; Hengl & MacMillan, 2019). Some of these covariates are typically derived from remote sensing data and digital elevation models, while machine learning algorithms are increasingly used for statistical modeling (Khaledian & Miller, 2020; Wadoux, Minasny & McBratney, 2020). For example, such an approach has been used to predict various soil properties at a spatial resolution of 250 m on a global scale (Hengl et al., 2017), for Africa (Hengl et al., 2015), for Argentina (Heuvelink et al., 2020) and for the managed forests of Canada (Mansuy et al., 2014), and at a 5 m resolution in a 580 km2 agricultural watershed in southwestern Burkina Faso (Forkuor et al., 2017).

Despite their well-demonstrated usefulness for predicting and mapping soil attributes (Viscarra Rossel et al., 2016), remote sensing data also have certain technical limitations (Barnes & Baker, 2000; Bartholomeus, Epema & Schaepman, 2007; Cécillon et al., 2009; Mulder et al., 2011; Hengl et al., 2017):

1. The covariates derived using remote sensing do not always cover the entire area to be mapped, so missing pixels must be filled using space-filling algorithms.

2. Vegetation, cloud cover, and other adverse weather conditions may hamper the accurate estimation of soil attributes from remote sensing data.

3. All covariates should be rasterized and scaled up or down to the desired resolution for predictions. This process may result in a loss of information in the scaled database.

4. The spectral signature of water surface and anthropogenic infrastructure such as urban areas, roads, buildings, power plants, airports, landfills, mining waste, etc., are not relevant to the mapping of soil properties. Therefore, these areas must be properly masked.

5. Because they only capture the properties of the earth’s surface, spectral signatures may be of little interest to map soils that present high vertical variability.

6. Temporal and spatial variation of soil properties such as soil moisture can reduce the accuracy of spectra-based models.

7. In forested areas, the effectiveness of using the forest cover’s spectral signature as an indicator of soil properties depends on indirect relationships between soils and vegetation. However, tree phenology, natural and anthropogenic disturbances, and forest dynamics induce spatial and temporal (seasonal to decadal) variations in forest cover. Therefore, they add noise to these relations. Forests are much more dynamic than soils, and short-term changes in forest cover do not necessarily translate into changes in soil properties. This is especially true for the boreal forest, where fire, insect infestations and logging are the main drivers of forest dynamics (Duchesne & Ouimet, 2008; Girard, Payette & Gagnon, 2008; Danneyrolles et al., 2019).

An alternative method to deriving environmental covariates from remote sensing data is to use field soil information in combination with a traditional (conventional) soil map (Hong et al., 2013; Hengl & MacMillan, 2019). Traditional soil (or ecoforest) maps are typically generated by manually delineating, interpreting, and classifying the shape and color of the surface and vegetation from stereoscopic multispectral aerial photographs in order to form map units with similar characteristics (Soil Science Division Staff, 2017). Soil characteristics are assumed to be relatively homogeneous within polygon boundaries. When available, expert interpretation of the land surface can serve to precisely define soil covariates over various areas of the landscape (Arrouays et al., 2014; Hengl & MacMillan, 2019). Field soil characterization and traditional soil maps are therefore one of the best sources of soil mapping information (Wiklander, 1975; Arrouays et al., 2014; Hengl & MacMillan, 2019).

In the province of Quebec, Canada, the ecoforest polygon map constitutes the basis of information for forest management (MFFP, 2020a). Over the past 50 years, the entire territory was analyzed every decade or so from black and white or infrared photographs (scale: ∼ 1:15,000). Polygons with common characteristics regarding forest attributes (composition, density, age, height), soil parent material, soil drainage, land slope, historical disturbances and ecological type have been delineated and characterized. Water bodies, farmland, unproductive land, roads and other non-forested areas are also delineated. The minimum area for the delimitation varies from 1 to 8 ha, depending on the delimited entity. Photo-interpretation follows standard protocols and is verified using a network of checkpoints that photo-interpreters visit to validate information. The information obtained is then rescaled. Full coverage (approximately up to the 52nd parallel) is publicly available in the form of 1:20,000 ecoforest maps (MFFP, 2020b).

To describe the existing forest resources in detail, the forest attributes of each polygon are also estimated from information compiled from forest inventory programs which are also run approximately every decade. As part of these surveys, several thousand soil samples have been collected and analyzed over the years to determine their texture. However, these laboratory analyses remain underutilized and, more importantly, they have never served to estimate the spatial variability of soil texture at the scale of ecoforest polygons. Using these precious data, the present analysis aims to map the texture of mineral soil at the scale of the ecoforest polygons in Quebec, Canada. This will allow a more accurate characterization of the spatial variability (2D) of soil texture which is currently roughly classified (fine, medium or coarse texture) based on photo-interpretation of soil characteristics for each ecoforest polygon. We hypothesised that soil particle size composition can be modelled and predicted from a set of environmental covariates, and that the most important model variables would be those observed in the field and those interpreted from aerial photography regarding soil characteristics.

Material and Methods

Study area

The study area corresponds to the forest area below the current northern limit of the managed forest in the province of Quebec, Canada. It extends up to approximately lat. 52°N and covers approximately 583,000 km2, of which 434,667 km2 are classified as productive forests. The normal mean annual temperature (1971–2000) varies approximately from −2.6 °C to 7.4 °C, and annual precipitation ranges from 770 mm to 1,600 mm (Duchesne et al., 2016). This territory comprises 3 different forest subzones which are mainly associated with variations in the mean annual temperature along the latitudinal gradient (Fig. 1). They are, from south to north: the hardwood forest, the mixed forest, and the continuous boreal forest (MRN, 2013). Based on available knowledge of the physiographic regions of Eastern Canada, 5 soil provinces also characterize the study area (Lamontagne & Nolin, 1997): the St. Lawrence Lowlands, the Appalachians, the Laurentians, the Abitibi and James Bay Lowlands, and the Mistassini Highlands (Fig. 1). They can be distinguished according to parent material, topography, and climate (Lamontagne & Nolin, 1997).

Figure 1 Forest subzones and soil provinces in Quebec, Canada.

Maps were produced with QGIS software, version 3.4 (QGIS, 2020). Basemap credit: ©2021 TerraMetrics, ©2021 Google, Esri, HERE, Garmin, ©OpenStreetMap contributors, and the GIS User Community.

Soil sampling and analysis

We used soil texture data from 29,570 mineral soil samples (17,901 sites) spread across the entire study area (Fig. 2). The data come from 3 different provincial forest inventory programs. The first program consists of more than 12,300 permanent sampling plots (PSPs, which are 400 m2 circular plots) gradually set up and inventoried by provincial forestry authorities since 1970 (MFFP, 2016). The PSPs are divided into 5 networks, each with specific objectives (MFFP, 2014). The basic network was installed first; it contains the largest number of plots (∼60% of the current total). The PSPs of this basic network are distributed randomly over the entire territory, with a sampling intensity of 1 plot per 26 km2 in the hardwood forest, 1 plot per 103 km2 in the mixed forest, and 1 plot per 259 km2 in the continuous boreal forest subzone. From 1989 to 1994, this basic network was supplemented by additional PSPs (∼25%) to increase the precision of forest growth models. The remaining PSPs (∼15%) are part of 3 networks deployed respectively to monitor the spread of insects and diseases, to obtain information on private forests and to document the effect of silvicultural treatments. Among the many variables recorded in PSPs, the first diagnostic B (thickness >10 cm) and C soil horizons (Soil Classification Working Group, 1998) were sampled for soil texture analysis 1 to 3 m outside the circular plot, after digging with a shovel and carefully observing the soil profile to select the desired horizons (MFFP, 2016). The relatively thin A horizon, generally characterized by the accumulation of organic matter or the by the eluviation of clay, soil organic matter, iron, or aluminum (Ae), was not sampled. A total of 19,255 soil texture analyses from 10,010 sites across Quebec’s managed forests come from the PSP forest inventory program.

Figure 2 Spatial distribution of mineral soil texture analysis ( n = 29,570) from 3 provincial forest inventory programs in Quebec, Canada.

PSP: permanent sampling plots; EOP: ecological observation plots; SIP: site index plots. The map was produced with QGIS software, version 3.4 (QGIS, 2020). Basemap credit: ©2021 TerraMetrics, ©2021 Google, Esri, HERE, Garmin, ©OpenStreetMap contributors, and the GIS User Community.

The second program is a major ecological inventory program that was carried out in Quebec’s forests at the end of the last century (MRN, 1994). A total of 28,425 ecological observation plots (EOPs, which are 400 m2 circular plots) were established from 1986 to 2000 to characterize the type of forest cover (composition, structure), understory vegetation (indicator plants) and soil characteristics (e.g., geologic deposit, drainage). Sampling intensity varies depending on the complexity of the terrain and vegetation, with approximately 1 plot per 15 km2 in the hardwood forest, 1 plot per 20 km2 in the mixed forest, and 1 plot per 25 km2 in the continuous boreal forest. The EOPs are distributed along transects that are positioned to cover the types of landform present in each ecological district and to probe specific features. The soil profile was completely characterized and the first diagnostic B and C sol horizons were sampled at a single point located within or near selected plots, with a frequency determined by the regional soil diversity (approximately one third of the EOPs; MRN, 1994). If the C horizon was absent or unreachable, the sample was taken as deep as possible in the soil profile. If the B horizon was absent or less than 10 cm thick, only the C horizon was sampled (MRN, 1994). This inventory program provides soil texture analyses for a total of 9,476 soil horizons (3,765 for horizon B and 5,711 for horizon C) from 7,085 sites across the managed forests of Quebec.

The third set of soil samples comes from the SIP program, a network set up to acquire data on forest productivity of the main ecological types encountered in the province of Quebec. It relies on tree disk sampling and stem analyses for site index determination (MRNF, 2008). At least 5 geographically well-distributed site index plots (SIP, circular 400 m2plots) were established in each of the 4 to 5 major ecological types found in each ecological region. Among the many records measured in each SIP, the first diagnostic B soil horizon (or C horizon if B is not present) was sampled for soil texture determination at a single point within or near the selected plots (MRNF, 2008). The SIP program provides soil texture data for 839 B horizons from 806 sites. For some unknown reason, 33 sites have 2 samples in the database. The fact that soil texture analyses of these duplicates are coherent for a given site suggests that these sites were sampled twice. Such duplicates in the dataset can also be the result of errors such as wrong attribution of soil samples to sampling locations or data entry errors. In any event, we chose not to arbitrarily discard these observations from the analyses.

In the laboratory, soil samples were air dried, crushed, and sieved through a two mm mesh sieve to exclude rock fragments. Particle size composition (sand [2-−0.05mm], silt [0.05–0.002 mm] and clay [<0.002 mm] fractions) was analyzed using the Bouyoucos hydrometer method (Bouyoucos, 1962). Samples were analyzed by external accredited laboratories before 2001, then by the organic and inorganic chemistry laboratory of the Direction de la recherche forestière.

Covariates

We compiled a set of 33 covariates including 22 variables related to climate, 5 related to soil characteristics, 3 to spatial position or spatial context, 2 to relief and topography, and 1 to vegetation (Table 1). To build the model, we selected covariates that had all been measured or observed in sampling plots of the 3 forest inventory programs along with climate data that were estimable for these plots according to their geographic location (see below). For mapping purposes, these covariates also had to have been characterized by photo-interpreters (or estimated for climate data) for all ecoforest map polygons for which we wanted to predict soil texture.

Table 1 List of environmental covariates used for soil texture modelling.

Category	Covariates	
Climate, annual averages (1981–2010)	
	1	Mean daily minimum temperature (°C).	
	2	Mean daily maximum temperature (°C).	
	3	Mean daily mean temperature (°C).	
	4	Mean daily mean temperature during growing season1 (°C).	
	5	Mean daily mean temperature in July (°C).	
	6	Total number of days without frost (days).	
	7	Longest period of consecutive days without frost (days).	
	8	Growing season lengt h1 (days).	
	9	Growing degree-day summation over 5 °C (°C).	
	10	Last frost day at spring (Julian day).	
	11	First frost day at falls (Julian day).	
	12	Aridity (mm)2.	
	13	Total precipitation (mm).	
	14	Total precipitation in June, July and August (mm).	
	15	Total precipitation during growing season1 (mm).	
	16	Total vapor pressure deficit (hPa).	
	17	Total vapor pressure deficit in June, July and August (mm).	
	18	Total Thornthwaite potential evapotranspiration (PET, mm).	
	19	Snowfall proportion (%).	
	20	Total snowfall (mm of water).	
	21	Total radiation (MJ/m2).	
	22	Total radiation during growing season1(MJ/m2).	
Soil characteristics	
	1	Soil horizon (diagnostic B or C).	
	2	Soil parent material: glacial without particular morphology (1A); glacial characterized by its morphology (1B); juxtaglacial (2A); proglacial (2B); fluvial (3); lacustrine (4); marine and littoral (5-6); thick organic (7E); thin organic (7T), slope and in situ weathered (8); eolian (9); bedrock (R, ≥ 50% of exposed bedrock).	
	3	Thickness of the soil parent material: thick (1: > 1 m); medium (2: ≥ 50 cm to 1 m); thin (3: ≥ 25 cm to 50 cm); very thin (4: < 25cm); thin to very thin (5: < 50 cm, ≥ 25% to < 50% of exposed bedrock); very thin or absent (6: ≥ 50% of exposed bedrock).	
	4	Soil drainage: excessively drained (0); somewhat excessively drained (1); well drained (2); moderately well drained (3); somewhat poorly drained (4); poorly drained (5); very poorly drained (6); complex (16).	
	5	Soil physical environment: based on a combination of synthetic soil texture (fine, medium, coarse) and synthetic soil moisture regime (subhydric, hydric, mesic, xeric) classes: Very thin (< 25 cm) mineral deposit, variously textured, xeric to hydric moisture regime or thin to thick mineral deposit, xeric to hydric moisture regime, very stony without matrix (0); Thin to thick mineral deposit, coarse texture, xeric or mesic moisture regime (1); Thin to thick mineral deposit, medium texture, mesic moisture regime (2); Thin to thick mineral deposit, fine texture, mesic moisture regime (3); Thin to thick mineral deposit, coarse texture, subhydric moisture regime (4); Thin to thick mineral deposit, medium texture, subhydric moisture regime (5); Thin to thick mineral deposit, fine texture, subhydric moisture regime (6); Thin to thick mineral deposit, hydric moisture regime, umbrotrophic (7); Thin to thick organic or mineral deposit, hydric moisture regime, minerotrophic (8); Thin to thick organic deposit, hydric moisture regime, umbrotrophic (9). (Organic soils were excluded from the modelling).	
Spatial context	
	1	Bioclimatic domains (n = 7).	
	2	Bioclimatic subdomains (n = 12).	
	3	Ecological regions (n = 46).	
Relief and topography	
	1	Altitude (m).	
	2	Slope: 0% to 3% (A); 4% to 8% (B); 9% to 15% (C); 16% to 30% (D); 31% to 40% (E); ≥41% (F); summit (S).	
Vegetation	
	1	Forest cover: deciduous (relative basal area of coniferous species < 25%); mixed (relative basal area of coniferous species between 25% and 74%); coniferous (relative basal area of coniferous species >75%); Non-forested (cover density <25%).	
Notes.

1 Growing season length is defined as the period (number of days) between the last 3 consecutive days with frost (daily minimum temperature <0 °C) in the spring and the first 3 consecutive days with frost in the fall.

2 Aridity is the accumulation of monthly water deficit (difference between monthly Thornthwaite potential evapotranspiration and monthly precipitation, zero if negative).

Climatic data include average annual values (1981–2010) of 22 variables. These covariates were simulated using the stochastic weather generator of the BioSIM software (Régnière et al., 2017). This model provides forecasts based on regional air temperature and precipitation, interpolated from nearby weather stations and adjusted for differences in elevation and location with regional gradients. Climate data were first estimated for each tile of a 0.5″ (∼1 km) resolution raster map, then extracted at the locations of sampling plots and ecoforest polygon centroids. For this, we used the raster package, version 3.0–12 (Hijmans, 2020) in version 3.5.1 of the R software environment (R Core Team, 2019).

The soil was characterized using the diagnosed soil horizon (B or C), soil physical environment classes, drainage classes, as well as the origin and depth of soil parent material. We included soil horizon (B or C) as a predictor variable because depth is unknown for PSPs and SIPs samples. In addition, having only 2 sampled depths does not allow the modelling of vertical variability with spline or parametric depth functions (Ma et al., 2021). We hypothesized that soil texture does not exhibit much variability between the B and C horizons at a given site, compared to inter-site variability, and focussed primarily on assessing the spatial variability of soil texture. The soil physical environment classes used in Quebec combine information from synthetic soil moisture regime and synthetic soil texture classes (Table 1; MFFP, 2016; MFFP, 2020a). We also characterized relief and topography using altitude and slope classes, and characterized forest cover using a simple 4-class system (deciduous, mixed, conifers, or regeneration). Finally, we captured variability associated with spatial position and context according to the bioclimatic domains, bioclimatic subdomains and ecological regions of the Ecological Land Classification Hierarchy (MRN, 2013; MFFP, 2020c). These spatial entities respectively divide the studied area into 7, 12 and 46 ecological units that represent various combinations of physical environments, climatic regimes, soil provinces, and vegetation. Therefore, these covariates potentially capture some of the spatial variability associated with these environmental characteristics. We included these variables, which reflect the spatial context, to account for the influence of environmental parameters that may have a local or regional influence on soil texture and would not be accounted for by the other covariates (e.g., geologic materials and land surface age since glacial retreat).

Statistical analysis and mapping

To ensure consistency between the predictions of sand, silt and clay fractions at each coordinate (which should total 100%), we first computed the isometric log ratio (ILR) to transform soil texture fractions with the compositions R package, version 1.40–4 (Van den Boogaart, 2020), and used the 2 functionally independent values (V1 and V2) for subsequent statistical modeling.

Sampling plot data and polygons of the ecoforest map were filtered to exclude agricultural and unproductive forest lands, organic soils (fen, bog), anthropogenic infrastructure, and water surfaces. Therefore, this study considers only productive forest land (defined as having the potential to produce more than 30 m3 of timber per hectare in 120 years or less) characterized by mineral soils.

We created dummy variables by converting all categorical variables to as many binary variables as there are categories, using the caret R package, version 6.0-85 (Kuhn, 2020). We then used tree-based random forest machine learning algorithms (method ranger from the caret package) to predict the V1 and V2 orthogonal components of soil texture considering all covariates in the analysis (133 covariates, including the converted dummy variables). To fine-tune the models, we also used the caret package to identify optimal values of the model tuning parameters based on the cross-validation performance. We used 5 repeats of 5-fold cross-validation, and tested a large range of tuning parameter values. The average root-mean-square error was used to select the optimal model using the smallest value. We also tested other machine learning algorithms, including gradient boosting, cubist, and k-nearest neighbors, but with our dataset, the random forest algorithm performed much better than these alternatives.

We evaluated the selected models by plotting observed vs. predicted values and comparing slope and intercept regression parameters against the 1:1 line (Piñeiro et al., 2008). We also computed the determination coefficient (R2), mean absolute error (MAE) and mean bias error (MBE) statistics using the postResample function of the caret package in the R programming environment (Willmott & Matsuura, 2005; Kuhn, 2020). This function calculates R2 by squaring the correlation between the observed and predicted values. We performed this evaluation on the V1 and V2 orthogonal components of soil texture and on the corresponding compositions (sand, silt and clay fractions) back-transformed from the modeled ILR-transformed values. Finally, we assessed the remaining spatial dependence structure of the model residuals by computing variograms of the cross-validation residuals using the gstat R package, version 2.0-4 (Pebesma, 2004; Gräler, Pebesma & Heuvelink, 2016).

After this parameterization, we used the models to predict the V1 and V2 orthogonal components of soil texture and the back-transformed particle size composition (sand, silt and clay fractions) for each ecoforest polygon of the provincial forest map. We also estimated the 95% prediction intervals for V1 and V2 using the quantile regression approach (Q.975–Q.025, Meinshausen, 2006; Vaysse & Lagacherie, 2017) using the ranger R package, version 0.12.1 (Wright & Ziegler, 2017). In order to reduce computing time, we produced provincial maps in the SIFORT mapping system to translate the conventional ecoforest polygon map (vector or object-oriented images) into a grid of tiles (mixed vector and raster images) separated by 15″ (∼375 m) (Pelletier, Dumont & Bédard, 2007). Each tile’s attributes correspond to the information for the polygon at the center of the tile on the conventional ecoforest map. This systematic sampling of the conventional ecoforest polygon map (∼7.7 million polygons) results in a relatively high definition raster map of Quebec’s forests (∼4.1 million tiles). In addition, to illustrate the variability of forecasts at finer spatial scales, we produced polygon maps at a chosen location using version 3.4 of QGIS software (QGIS, 2020). Particle size composition (sand, silt and clay fractions) was presented on a ternary color scale where the hexadecimal RGB codes from ternary compositions were computed with the tricolore R package, version 1.2.2 (Schöley & Kashnitsky, 2020). We used the dplyr R package, version 0.8.3 (Wickham et al., 2019) for data manipulation as well as the ggplot2 R package, version 3.3.0 (Wickham, 2016) and the cowplot R package, version 1.0.0 (Wilke, 2019) for graphic production.

Results

Model performance

The observed vs. predicted values of the V1 and V2 orthogonal components of soil texture and the corresponding particle size composition (sand, silt, and clay fractions) are presented in Figs. 3 and 4, respectively. The models explain 46% and 57% of the variance (R2) for both orthogonal components, with mean absolute errors of 0.39 and 0.41 and mean bias errors of ±0.001 (Fig. 3). For both models, slope and intercept parameters of the linear regression between observed and predicted values are very close to the 1:1 line. After back transformation, these models explain 60%, 34%, and 78% of the variance in sand, silt and clay fractions, respectively, with mean absolute errors ranging from 4.0% for the clay fraction to 9.5% for the sand fraction, and mean bias errors ranging from −1.0% to 1.2% (Fig. 4).

Figure 3 Observed vs. predicted values of the V1 and V2 orthogonal components of each soil sample’s particle size composition (sand, silt and clay fractions).

The straight blue line corresponds to the linear regression between observed and predicted values, and dotted lines represent the 1:1 line. The 0 values of observed V1 that are aligned horizontally in the top graph correspond to observations with an identical composition of sand and silt or to soils composed entirely of sand (100%), while those of observed V2 in the bottom graph correspond to soils characterized by 0% clay. The lower rows of horizontally aligned V1 values (top graph) correspond to integer sand values of sandy soils ( >90%).

Figure 4 Observed soil texture composition (sand, silt and clay fractions) vs. predicted values, back-transformed from ILR-transform (V1 and V2) predictions.

The straight blue line corresponds to the linear regression between observed and predicted values, and dotted lines represent the 1:1 line.

Covariate importance

The measures of relative covariate importance in the V1 and V2 orthogonal component models are presented in Fig. 5. The most important variables are those observed in the field and interpreted from aerial photography regarding soil characteristics (soil physical environment class and soil parent material, Table 1), followed by those pertaining to elevation and climate, the latter all having comparable importance. Spatial context (Ecological Land Classification Hierarchy), terrain slope, soil horizon (B or C), and forest cover also contribute to the explained variance, but to a lesser degree.

Figure 5 Relative measures of variable importance in the selected models of the two orthogonal components (V1: left panel; V2: right panel) of soil texture composition (sand, silt and clay fractions).

Dot color indicates variable category. Only the 50 most important variables are shown. See Table 1 for covariate definitions.

Spatial dependence structure

Variograms illustrating the spatial dependence structure of the V1 and V2 orthogonal components of particle size composition (sand, silt and clay fractions) and of the model’s residuals are shown in Fig. 6. The fact that all 4 variables were found to fit spherical models with a nugget effect denotes spatial structure in the data. Spatial structure was observed over a range of 14.9 to 16.3 km. Spatial dependence between data can be quantified using the nugget-to-sill ratio (NSR, Cambardella et al., 1994; Aidoo et al., 2015). Models captured part of the spatial autocorrelation, as denoted by the greater NSR ratio for residuals than for observations. Nevertheless, a moderate degree of spatial autocorrelation remains in the residuals.

Figure 6 Variograms illustrating the spatial dependence structure of the V1 and V2 orthogonal components of observed soil particle size composition of the B horizon and of the model’s residuals.

The lower dotted horizontal lines represent nuggets (y-axis intercept-related amount of short-range variability in the data) and the upper lines represent the sills (total variance at which the model first flattens out). Vertical dotted lines represent the range (distance beyond which data are no longer spatially correlated). NSR: nugget-to-sill ratio.

Mapping

Figure 7 presents the gridded map (15″ resolution) of soil particle size composition in ecoforest polygons for the area of Quebec’s managed forest. At the provincial scale, we visually distinguish the Abitibi and James Bay Lowlands soil province in the northwest of the study area. It is characterized by flat topography, with organic deposits in lowland areas in its northwestern part, and mostly fine-textured lacustrine or marine deposits at higher altitudes in its eastern and the southern parts (Blouin & Berger, 2005). The fine-textured mineral deposits originated from the proglacial Ojibway lake in its southern part, and from the prehistoric Tyrell sea that existed during the retreat of the North American ice sheet in its northern part.

Figure 7 Gridded map (15 s resolution) of soil texture composition (diagnostic B horizon) in ecoforest polygons of the managed forest of the province of Quebec, Canada.

Only productive forest land characterized by mineral soils was mapped. Agricultural and unproductive forest land, organic soils, anthropogenic infrastructures, and water areas were excluded. Hillshade effect was added based on a 1km resolution digital elevation model. The map was produced with QGIS software, version 3.4 (QGIS, 2020). Basemap credit: ©2021 TerraMetrics, ©2021 Google.

Maps also highlight the medium-textured soils of the Appalachian soil province, located south of the St. Lawrence River. This soil province is characterized by ridges of flattened summits, rocky crests, undulating hills and valleys. The main parent materials are glacial tills on mountaintops and slopes, fluvial deposits along streams, and organic deposits in depressions.

Coarser-textured soils are observed in the Laurentians and the Mistassini Highlands soil provinces, both located north of the St. Lawrence River and east of the Abitibi and James Bay Lowlands soil province. The landscape of these two soil provinces consists of ridged rolling hills, crests, and valleys; the main types of parent materials are sandy or gravelly glacial till on hills, sandy or gravelly fluvial deposits in valleys and along water bodies, and organic deposits in depressions.

Finally, the map also highlights the coarse-textured soils of the St. Lawrence Lowlands soil province in southern Quebec. Marine deposits in this soil province are from the prehistoric Goldthwait (southern Quebec) and Laflamme (Lac Saint-Jean area) seas. Fine-textured soils predominate at lower altitudes, while coarse-textured soil deposits are more abundant at higher altitudes. However, the mapped forest soils in this soil province are mainly coarse-textured; this can be explained by the fact that fine-textured soils are predominantly farmlands (which were not mapped), and by the much higher density of anthropogenic infrastructure in this region.

As examples, we also produced maps illustrating particle size composition of mineral soils in ecoforest polygons at scales of 1:200,000 and 1:100,000, for a region located at the southern edge of the Abitibi and James Bay Lowlands soil province, approximately 25 km south of Lebel-sur-Quévillon (Fig. 8). This region shows a transition from low altitudinal fine-textured and sandy juxtaglacial and lacustrine deposits originating from the prehistoric Lake Ojibway in the west, to medium- and coarse-textured soils in the east, where somewhat thicker till deposits dominate. Organic soils, for which mineral soil texture cannot be estimated, are also abundant in this region.

Figure 8 Maps of soil texture composition at two zoom levels, for a region at the southern edge of the Abitibi and James Bay Lowlands soil province.

Only productive forest land characterized by mineral soils was mapped. Agricultural and unproductive forest land, organic soils, anthropogenic infrastructures, and water areas were excluded. Maps were produced with QGIS software, version 3.4 (QGIS, 2020). Basemap credit: ©2021 TerraMetrics, ©2021 Google.

Mapping accuracy

Evaluation of the 95% prediction intervals of the V1 and V2 orthogonal components of soil texture reveals that the estimates of the particle size composition are more precise for mineral soils with coarse texture, and less precise for those with fine texture (Fig. 9). This results in regional differences in mapping accuracy, as estimates of soil texture are less accurate in regions dominated by fine-textured mineral deposits than in regions primarily characterized by coarse-textured mineral deposits. At a finer spatial scale, this also translates into a difference in forecast accuracy between soil physical environment classes and soil parent material types, which were the 2 most important variables in the models.

Figure 9 Gridded map (15 s resolution) of the 95% prediction intervals of the V1 and V2 orthogonal components of soil texture composition (B horizon) in ecoforest polygons of the province of Quebec.

Only productive forest land characterized by mineral soils was mapped. Agricultural and unproductive forest land, organic soils, anthropogenic infrastructures, and water areas were excluded. Maps were produced with QGIS software, version 3.4 (QGIS, 2020). Basemap credit: ©2021 TerraMetrics, ©2021 Google.

Discussion

Model performance

From a set of 33 covariates (133, if we include the converted dummy variables), our models respectively explain 60%, 34%, and 78% of the variance (MAE = 9.6%, 7.9%, 4.0%) in sand, silt and clay fractions of 29,093 soil samples distributed throughout the managed forest of the province of Quebec, Canada. By comparison, Hengl et al. (2017) explained 73% to 79% of these soil characteristics (MAE = 6.6% to 9.0%) from a set of 158 remote sensing-based soil covariates for more than 600,000 globally distributed soil samples (about 150,000 soil profiles). More precisely, we explained a smaller proportion of the variance in the silt fraction, but we modeled clay fraction with higher precision (smaller error). In contrast, Mansuy et al. (2014) explained only 20%, 13% and 43% of the variance in sand, silt, and clay fractions of 538 sample plots distributed throughout the Canadian managed forest from a set of 12 topographic and climatic soil covariates. Their forecast was also much less precise than our results and than the global models of Hengl et al. (2017).

A closer look at the SoilGrids map of clay content (https://soilgrids.org/, Hengl et al., 2017) reveals that global predictions completely fail to characterize the fine-textured soils of the Abitibi and James Bay lowlands soil province in Quebec, Canada (Fig. 7), probably due to the very low number of soil samples from this region. Also, as compared to our polygon map, global and Canadian forecasts of soil texture were provided for mineral as well as for organic soils, apparently due to the inability to discriminate organic deposits in the mapping exercise. In addition, despite the higher spatial resolution of the SoilGrids and Canadian maps (250 m), rasterized predictions are somewhat smoothed over the territory. By comparison our polygon map shows much more spatial variability in terms of particle size distribution (Figs. 7–8). The rendering of these abrupt spatial transitions between polygons is made possible by the characterization and delineation of soils by photo-interpreters. Since photo-interpretation also facilitates the determination of land usage at a relatively fine spatial scale, our polygon map provides forecasts for mineral soils from an entire region in southern Quebec characterized by Mansuy et al. (2014) as agricultural land (and therefore, not mapped), while excluding land with anthropogenic infrastructure.

The cross-validation procedure, which involves splitting the original dataset repeatedly into calibration and validation datasets, is commonly used to validate predictive models of soil characteristics, because collecting additional independent samples is often impractical (Hengl & MacMillan, 2019). This procedure does not rely on data that is independent of the original sampling design. Thus, if the sampling is biased or unrepresentative, the validation procedure may not reveal the model’s true accuracy. However, if the sampling is unbiased—as we are confident was the case in this study—the randomly selected subsets provide unbiased estimators of the model’s true accuracy (Hengl & MacMillan, 2019).

Variable importance

The most important variables in our soil texture models are soil physical environment classes and soil parent material, followed by altitude and variables regarding climate, which all have a comparable importance (Fig. 5, Table 1). Most important rasterized covariates used for the Canadian soil texture maps by Mansuy et al. (2014) are 4 climate variables (3 variables related to air temperature and 1 to precipitation) and 4 topographic variables (aspect, slope, elevation, and watershed stream). In Hengl et al. (2017) global models, the most important covariate to predict soil texture, by far, is depth (up to 2 m) below the soil surface. This suggests that a large part of the explained variance of the ∼600,000 soil samples stems from the vertical variability of observations among the ∼150,000 soil profiles. In contrast, our evaluation of model performance and variable importance mainly refers to the spatial variability of soil texture over the studied area. Indeed, we used only 1 or 2 samples per plot (29,570 mineral soil samples from 17,901 sites) for model parameterization.

We hypothesized that soil texture does not vary much between the B and C horizons at a given site, compared to inter-site variability. Thus, we focussed primarily on assessing the spatial variability of soil texture. Nevertheless, we included soil horizon (B or C) in the modelling exercise to take into account the variability of soil texture associated with soil horizon and, indirectly, with sampling depth. Our analyses of each variable’s importance revealed that horizon does not rank among the most important variables in our models (Fig. 5). This in itself provides evidence that the vertical variability in soil texture is small compared to spatial variability. In addition, tests for association revealed that the clay fractions of paired soil samples of B and C horizons were closely correlated (Pearson’s product moment correlation coefficient of 0.86, t = 183, df = 11 996, p-value <0.001).

After depth below the soil surface, the most important variables in global models of soil texture are mean monthly precipitation, mean monthly temperatures, monthly MODIS (Moderate Resolution Imaging Spectrometer) perceptible water vapor images, and digital elevation model-parameters (Hengl et al., 2017). The great importance of climatic variables in global models translates into smoothed forecasts at a local scale. Our results confirm that climate and remote sensing covariates are less relevant for soil texture mapping than the characterization and delineation of soil characteristics by trained photo-interpreters. For example, soil physical environment class 6 (40% clay on average) and 3 (30% clay on average) as well as class 2 (8% clay on average) and class 1 (5% clay on average) are among the most important variables in the models (Table 1, Fig. 5). Similarly, fine-textured lacustrine deposits (41% clay on average), coarse-textured glacial deposits without particular morphology (8% clay on average) and proglacial deposits (4% clay on average) are also among the most important discriminating variables of soil texture (Table 1, Fig. 5).

Spatial dependence structure

Although our models explain much of the spatial variation in the data, moderate spatial autocorrelations remain in the model residuals. Regionally, spatial interpolation (kriging) of residuals might improve predictions. Hybrid regression-kriging models apparently performed better than individual models for digital mapping of soil organic carbon (Lamichhanea, Kumara & Wilsona, 2019). However, combining kriging of residuals with random forest predictions did not always outperform random forest predictions alone (Lamichhanea, Kumara & Wilsona, 2019). As suggested for global soil maps, kriging of residuals would only marginally improve forecasts at the provincial scale, and would come at significant computing cost (Hengl et al., 2017). Combining kriging of residuals with random forest predictions also raises issues related to prediction error propagation. The accuracy gains that could result from kriging of residuals as well as the methodology to be used for propagation of uncertainty remain to be established.

Hengl et al. (2018) proposed another framework for spatial prediction with random forest algorithms, in which buffer distances from sample coordinates are used as explanatory variables to incorporate spatial structure into the modeling process. Regionally, it could also be used to improve predictions; however, it is not adapted for the analysis of large soil sample datasets as the one used in the present study (Hengl et al., 2018).

Mapping accuracy

The assessment of 95% prediction intervals allowed us to characterize the spatial distribution of uncertainties associated with the orthogonal components of particle size composition estimated from a set of environmental covariates. The comparison of observed and predicted values (Fig. 4) and the maps of the prediction intervals (Fig. 9) indicate that soils containing more than about 25% of clay are less frequent, and that their particle size composition (up to 80% clay) is more difficult to estimate accurately. Thus, models are more imprecise for fine-particle soils characterized by high clay content.

Many sources of uncertainty can affect the overall accuracy of mapping, including field sampling, data entry errors, laboratory analysis, characterization of covariates at sampling sites and by photo-interpreters, polygon delineations, climatic covariate estimations, short-range variability, etc. (Hengl & MacMillan, 2019). Our prediction interval maps only refer to the accuracy of the mathematical model; they do not reflect the imprecision of the photo-interpreted covariates, which remains poorly documented. Better quantification of the accuracy of the expert interpretation of land surface and of covariate characterization by photo-interpreters at targeted sites would allow more accurate mapping of the cumulative error associated with soil texture predictions (Kempen et al., 2010). However, this shortcoming only applies when prediction accuracy is evaluated at locations where field data are not available. It does not affect the evaluation of model performance and of prediction uncertainty at sites where covariates have been characterized by field observers.

Soil characteristics are known to be difficult to estimate with precision. Our estimates compare favorably to other similar exercises, particularly thanks to the high sampling intensity, the availability of expert interpretation of the land surface, and the delineation of ecoforest polygons from aerial photography for the entire territory studied. However, the approach used in this study cannot easily be extrapolated to other territories where such information is not available. Also, the propagation of the prediction error associated with V1 and V2 orthogonal components of soil texture to the back-transformed particle size composition (sand, silt and clay fractions) is a challenge that requires further research.

Conclusion

We used soil texture data from 29,570 mineral soil samples from 17,901 soil profiles to model and map particle size composition in ecoforest polygons of the provincial ecoforest map of Quebec, Canada, using a set of 33 covariates. Our results compare favorably with previous soil texture mapping studies for the same territory, in which soil texture was modeled mainly from rasterized climatic and topographic covariates. Relative measures of variable importance in the models confirm that the characterization and delineation of soil characteristics by trained photo-interpreters are more relevant for soil texture mapping than climate and remote sensing covariates. Depending on the needs, additional rasterized environmental covariates and methods that incorporate spatial structure into the modeling process could be considered to improve predictions at a local scale. Nevertheless, with mean absolute prediction errors ranging from 4.0% for the clay fraction to 9.5% for the sand fraction, the map we provide should meet the needs of provincial forest managers, as it is compatible with the ecoforest map that constitutes the basis of information for forest management in the province. It could also help refine the contours of soil provinces and subregions, which have only been roughly delineated on a large scale and only as far north as 51°N, based on available knowledge of the physiographic regions of Eastern Canada.

Supplemental Information

Supplemental Information 1 Raw data used for soil texture modelling in ecoforest polygons in Quebec, Canada

Click here for additional data file.

We wish to thank the staff of the Direction des inventaires forestiers of the MFFP, who managed the forest inventory program and supervised all field contractors who carried out soil sampling over the years. We also acknowledge the personnel of the organic and inorganic chemistry laboratory of the Direction de la recherche forestière for performing numerous textural analyses, Marie-Claude Lambert who generated meteorological data, Jean Noël for assistance in spatial data processing and Denise Tousignant for English editing. We are also grateful to the editor and the three anonymous reviewers for their valuable comments and suggestions to improve the quality of the paper.

Additional Information and Declarations

Competing Interests

Author Contributions

Data Availability

The authors declare there are no competing interests.

Louis Duchesne conceived and designed the experiments, performed the experiments, analyzed the data, prepared figures and/or tables, authored or reviewed drafts of the paper, and approved the final draft.

Rock Ouimet conceived and designed the experiments, authored or reviewed drafts of the paper, and approved the final draft.

The following information was supplied regarding data availability:

The raw data used for soil texture modelling are available in the Supplemental File.

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
