# Peer review of "Digital mapping of soil texture in ecoforest polygons in Quebec, Canada"

_PeerJ, doi:10.7717/peerj.11685_

## Round 0.1 · original submission · Major Revisions

The manuscript addresses an interesting subject and is well written. However, all three reviewers recommend major revisions and I agree with them. The reviewers have done thorough reviews, which can help the authors improve the manuscript. Please tackle all reviewers’ suggestions. Only for the recommendation of reviewer 3 of adding a flowchart, I think this is optional.

I agree with reviewer 1 that the objective of the study and the knowledge gap to be filled should be clearly established.

Besides the reviewers’ recommendations, I have some additional comments:

Materials and Methods
Study area
I suggest adding a map showing the forest subzones and soil provinces.
Soil sampling and analysis
L 156-157: Were A horizons not sampled? Why?
L 182: Please specify the particle size range for each fraction
Covariates and Table 1
Here are my main concerns:
1. Under the heading "Parent material", several variables that are not parent material are included: Soil horizon is not parent material. Soil thickness or stoniness is not parent material. Soil drainage is not parent material. What you call "soil surface deposit" seems to be parent material.
2. Which is the difference between "in situ weathered" and "bedrock"?
3. Does “Thickness of soil deposits” refer to the soil thickness or to the thickness of the sediment which constitutes the soil parent material?
4. Soil physical environment: "Xeric" and "mesic" are not types of drainage. Xeric is a moisture regime (related to precipitation and evapotranspiration), mesic is a temperature regime. Please reword or explain what you mean by "xeric to mesic drainage". Moreover, you have used "Soil drainage" in point 4. Please define "hygric" or “hydric” drainage. In addition, you include here "coarse texture", "medium texture" and "fine texture". How can these be covariates influencing soil texture?
5. As for “organic (8: Fen, 9: bog)”, according to the section Material and Methods, organic soils (fen, bog) have been excluded.
L 221 and captions of Figures 6, 7 and 8: does "mineral soil deposits" mean "mineral soils"? (soils can be developed from deposits but are not deposits).

Results
Covariate importance
L 270-271: In what you call "soil physical environment" you include (Table 1) "coarse texture", "medium texture" and "fine texture". It is not surprising that these variables had a great importance in soil texture. The categories included in "soil physical environment" must be clearly defined.

Discussion
Variable importance
L 358-359: Please reconsider what you call "soil physical environment" and "soil surface deposits". The first is not understandable and includes soil texture (the dependent variable). The second is apparently soil parent material.
L 369-370: Does this mean that soil texture is quite homogeneous in depth or that for most soils you have used only one depth?
L 375: Does “locally” mean in your study?
L 375-377: Is this because climate varies much less than soil parent material within the studied area?
I reiterate that "soil physical environment" must be reformulated. My interpretation is that the greatest importance is attributable to the soil parent material, which you curiously call “soil surface deposit”.
It should be highlighted here which parent materials gave rise to fine, medium and coarse textured soils.

Reviewer 1 ·

Basic reporting

The paper is clearly written. I have provided some editorial comments in the general comments section.

The paper is adequately referenced

The figures and tables are fine with the exception of points that I have provided in the general comments section.

No hypotheses were presented in this paper.

Experimental design

The paper describes original primary research and appears to be within the aims and scope of the journal.

The research question in this paper does not seem to be clearly stated. It is unclear how the research fills a knowledge gap. The knowledge gap has not been clearly identified. I think that the authors should be able to more clearly state their research question (s) and identify knowledge gap (s)

The authors appear to have performed a rigorous investigation to a high technical and ethical standard.

The methods are described in sufficient detail. Some comments are provided in the general comments section.

Validity of the findings

The study appears statistically sound.

Lines 404-414. The conclusions from the study seem fairly limited. This may be related to the relatively limited aim of the study “to map the texture of mineral soil at the scale of the ecoforest polygons”. The authors should consider ways to make the study appear more than a relatively straightforward mapping exercise.

Additional comments

The authors state on lines 126-127 that “the present analysis aims to map the texture of mineral soil at the scale of the ecoforest polygons in Quebec, Canada.” The authors used an impressive dataset of 29570 mineral soil samples (17901 sites). However, the authors describe their study as an exercise in mapping soil texture in Quebec. It would be good if they provided research questions, objectives or hypotheses and clearly stated the research gap that they are attempting to fill.

Line 187. The authors state that they compiled 33 covariates (and again on line 331), however, the authors should indicate how many covariates they actually used in the modelling. The number appears to have been much higher due to the creation of dummy variables as described on line 222. Figure 4 includes “the 50 most important variables”, suggesting there were many more than 33 covariates used.

Lines 190-192. The authors need to clarify in detail what they mean by “estimable”. They need to state exactly how they obtained the covariate data.

Line 201 It is not clear what the authors mean by geological material. This is not the correct terminology.

Lines 186-211. The authors need to clearly state the sources of their covariate data. For example, where exactly is the “soil physical environment” data mentioned in line 202 from. This needs to be clarified. Also, the term “soil physical environment” is potentially misleading and should be changed to something such as “soil texture and drainage class”.
The authors refer to “soil physical environment” in line 202-203 and in Table 1 they describe the covariates used as “Soil physical environment: very shallow (< 25 cm) or very stony (0); coarse texture and xeric to mesic drainage (1); medium texture and xeric to mesic drainage (2); fine texture and xeric to mesic drainage (3); coarse texture and hygric drainage (4); medium texture and hygric drainage (5); fine texture and hygric drainage (6); hydric drainage (7); organic (8: Fen, 9: bog).” Was this information collected in the field or obtained from another source? This needs to be clarified. It appears that possibly, the information was included in the ecoforest polygon mapping, but this needs to be clarified.

Lines 358 and on. It appears that the authors used mapped texture class (coarse, medium, fine) information as a covariate in their modelling. They then found that the most important variables in their models were variables “Describing soil physical environment and soil surface deposits” (line 359). In particular, SoilPhysEnv 1 and 6 as indicated in Figure 4 were found to be important and these are” coarse texture and xeric to mesic drainage (1); fine texture and hygric drainage (6)” as indicated in Table 1. This seems to suggest that the authors used estimates of soil textural class from the ecoforest polygon mapping as a covariate for their modelling. Essentially, it seems that maps of soil texture in categories of coarse, medium and fine texture could be produced directly from the ecoforest polygon maps prior to this study, and that what the authors have done is refined this mapping to produce maps of soil texture with more detail. This should be clearly stated and discussed in the paper (introduction, methods, results, discussion, conclusion).

Line 373-374 A reference is needed here.
Figure 1. there should be some labels on the map to orient the reader, such as the border of Quebec, location of a few major cities, and names of water bodies

Minor Editorial Comments
Line 25 “on the field” should be “in the field”
Line 78. The point is not stated as a technical limitation.
Line 155 “sylvicultural” should be “silvicultural”
Line 165 “”the mixed forest” should be “in the mixed forest” and “the continuous boreal” should be “in the continuous boreal”
Line 197 “tiles” should be “tile”
Line 214 “totalize” is not a word
Line 230 “others” should be “other”
Line 354 “from entire region” should be “from an entire region”
Line 377 What is meant by “experimented photo-interpreters”?
Line 387 “a large soil sample datasets” should be “large soil sample datasets”
Line 413-414 The meaning of the last sentence in the conclusion is unclear.

Reviewer 2 ·

Basic reporting

See General comments for the author below.

Experimental design

See General comments for the author below.

Validity of the findings

See General comments for the author below.

Additional comments

This is a well-written and well-structured manuscript that applies machine learning to map soil texture in a large ecoforest region in Quebec, Canada. The approach is fairly standard but it is state-of-the-art and has some novel elements too. The methodology is overall sound but not always clearly explained and may be criticised on some aspects. My concerns are worked out in the detailed comments below.


Detailed comments

(L97-107) It is not about whether a vector or raster representation of covariates is used, these are just technical aspects that do not truly influence the statistical modelling. In the end it is about how much of the soil variation is explained by the covariates, regardless of whether they are polygon or raster maps. What is important is whether soil variation is considered abrupt or gradual. Authors assume that soil spatial variation within polygons is small compared to variation between polygons, but is this true? This is just an assumption as stated in L102, so I don’t think you can maintain the general claim in L105-107.

(L108-119) Clearly a lot of effort has gone over the past 50 years into producing the ecoforest polygon map and indeed it turns out to be a valuable source of information in this study. That in itself is very fine but note that it is not a solution in cases where such information is not available. In other words, the approach used in this work cannot easily be extrapolated to other studies. Perhaps this could be mentioned in the Discussion. Authors might even check whether detailed polygon maps are available for other parts of the world and in this way discuss the extrapolation potential of their method. It was also not clear to me how the polygon maps were used in this study. Were only the polygon delineations used, by averaging covariates within polygons (L191-192 seems to suggest this, it is also apparent from Figure 7 that shows abrupt changes in texture predictions at polygon boundaries)? Or were the attribute values of the polygons (L111-113) used as covariates? I did not see these mentioned clearly in Table 1, so perhaps they weren’t? If not, why not? If they were used, there may be a problem that the soil observations used for calibration and cross-validation were also used to create covariates, which leads to overoptimistic validation metrics.

(L135-136) I did not come across the term ‘soil province’ before, is this a common term? If not, please define it.

(L143) There are more soil samples than sites, is that because sometimes there more than one observation per profile or is that because there are multiple locations per site. I think it is the former, because later on authors explain that samples from the B and C horizon were taken, but here I was not sure. Perhaps it is because of the term ‘site’ which is larger than a point location. For the PSPs the sites are 400 m2 circular plots (L146), so how was the soil sampling done? Were these soil samples from the centre of the circle or composite samples from multiple points within the plot? This should be clarified, also for the other two datasets (EOPs and SIP). Since soil data from three different programs were merged, it is important to describe any possible systematic differences between field and lab work between the programs, and whether this may have affected the results. L181-185 indicates that all soil samples were treated exactly the same way, over a very long period of time. Is this true?

(L156, L169-170, L178) Soil samples were taken from the first diagnostic B and C horizon. These were all merged into a single dataset but in the modelling the horizon was included as a covariate (Figure 4, Table 1). So when the calibrated model is used for prediction we may predict texture both for the B and C horizon. Figures 6 and 8 refer to the B horizon (Figure 7 presumably too, but this is not specified in the caption). What about the texture maps of the C horizon? And how to interpret the B horizon texture map for locations that have no B horizon (this happens, see L178-179)? Note that fitting a single model that assumes that a single binary covariate “B or C horizon” explains the difference between texture in the B and C horizon is quite rigid, what about fitting separate models for the B and C horizons? Was this tried and if yes how well did it predict soil texture? I would also like to see summary statistics of the top, bottom and thickness for both horizons, how variable are they? In the end we get maps that predict soil texture for horizons, but for users such as provincial forest managers, such maps may only be useful if it is also known what the depth intervals of the B and C horizon are at every prediction location. Are the B and C horizon indeed most interesting for forest managers, more so than the A horizon? Can they reliably be sample using an auger (L156)? Why was it done this way instead of mapping soil texture at fixed depth intervals (this is what most studies do, including the Hengl. et al (2017) study that authors use for comparison)? Soil samples were taken in one-third of the EOPs, was this subset chosen randomly from all EOPs?

(L171) Only 9476 horizons of 7085 sites were sampled. Why not 2 x 7085? Is this because for many sites there was no diagnostic B horizon? But this would mean that diagnostic B horizons are quite rare? Is this not a problem?

(L178-180) The text suggests that for each site no more than one soil sample was taken, but we still have 839>806.

(L201-203) This suggests that geological covariates were measured at the sampling sites, but these are not known at prediction locations (unless they are interpolated in some way, but this introduces uncertainty) so cannot be used to calibrate the machine learning model. Please clarify. It may be that all covariates were averaged over the polygons before they were offered to the machine learning model, but this was not clear to me. If this was done, then state this more clearly. Also, why would this be a proper way of preparing covariates? It is built on the assumption that soil and covariate variation within polygons is negligibly small, but is that really the case? Why would it be needed, why not use the covariates as raster maps? Maybe I simply did not get it, but in a way this then indicates that authors can do a better job in explaining how the polygon approach was applied and why it is useful.

(L205-209) Why refer to ecological regions based on physical environment, climate regimes, soil provinces and vegetation as ‘spatial position and context’? Are these not covariates that refer to environmental conditions (i.e., climate, vegetation, terrain)?

(L228-231) The type of machine learning algorithm and the machine learning hyperparameters were also optimised. What dataset was used for that? To guarantee independence, the data used for this should not also be used for model performance evaluation. In other words, a nested cross-validation approach should be used, where data are split in three parts (calibration, validation and test data).

(L234) Does the R-square refer to that computed on the 1:1 line or on a linear regression of predicted on observed (i.e., the squared Pearson correlation)? This should be clarified, and in fact the first of these should be used. To avoid confusion it would be better to refer to it as ‘model efficiency coefficient’ (for example see https://en.wikipedia.org/wiki/Nash%E2%80%93Sutcliffe_model_efficiency_coefficient).

(L243, L321-328, L389-402) Authors quantified prediction uncertainty of V1 and V2 but what we really want is the uncertainty of the clay, silt and sand predictions. Perhaps they could try and compute these. This requires that the correlation between the V1 and V2 prediction uncertainty is quantified (can be done, see https://arxiv.org/abs/2005.14458) and next that an uncertainty propagation analysis is done (Monte Carlo simulation is easy but computationally demanding, perhaps it can be done analytically or using a Taylor series approximation approach). Perhaps it is too challenging, but if they succeed it would be a very worthwhile extension of the methodology. If not feasible, it may be mentioned as a useful topic for future research in the Discussion. As it is now, Figure 8 and L321-328 present and discuss the V1 and V2 uncertainty maps but can these be easily interpreted in terms of texture, given that the transformation from V1 and V2 to clay, silt and sand is highly non-linear?

(L245-251) Not very clear what was done here, it suggests that in the end predictions were made on a grid after all. So where does the polygon approach go? Was that only used to average covariates within polygons (which may be criticised, see comment above)? And what are ‘regional polygon maps’? How do these differ from the ecoforest polygon map introduced in L108?

(L279-280) That depends on the nugget-to-sill ratio: if this is small, I would not conclude that there is a high amount of short-range variability. Please reformulate.

(L287-319) I like that authors pay ample attention to interpretation of the resulting maps and explain the patterns that were obtained from a pedological perspective.

(L333-341) Interpretation and comparison of the proportion of variance explained is tricky for study areas that are not the same because it is a ratio and depends on the amount of variance present in the study area. For example, for global mapping as done in Hengl et al. (2017) it is relatively easy to get a high R-square because the denominator of the ratio may be larger for global data than for local/regional data. A more honest comparison would be using the RMSE or MAE.

(L350-356) The abrupt spatial transitions in Figures 6 and 7 is simply the result of the modelling approach used here (if I am right authors averaged covariates within polygons before using them to train the machine learning model). The text suggests that this is a good thing but it is merely a result from the choices made, and it was not shown that his is better than alternative approaches that do not pre-process covariates. Authors should prove that their approach is better by comparing it with a method that trains a machine learning model on the same covariates, but then without first averaging within polygons, using the same training data.

(L383) It seems to me that the computing costs of residual kriging would not be that demanding in this study, if at least authors would limit the conditioning data to data within a local neighbourhood. Why not do this and check out the improvement?

(L410-411) That really also depends on the accuracy of the map. Different uses require different accuracy levels, so substantiate this claim by quantifying the uncertainty of the prediction maps and evaluating if these are within limits required by provincial forest managers.

(Figure 2). Explain the horizontal bands in these figures. The straight blue line supposedly represents the linear regression but it is very close to the 1:1 line. I find this hard to believe. It should have a much steeper slope (because when models explain only part of the spatial variation, predictions are smoother (have lower variance) than observations. Same problem occurs (but to a lesser extent) in Figure 3.

(Figure 5) Is it realistic to assume that residuals in the B and C horizons have the same variogram? This goes back to the fairly rigid assumption that largely the same model may be used for both horizons (same holds for Figure 4, also here the importance of covariates is forced to be te same for the B and C horizons). Perhaps more accurate models can be obtained if a more flexible model was used (separate models for B and C horizons).

(Figure 7) It may be useful to indicate this area by a rectangle in Figure 6. What do scale 1:200,000 and scale 1:100,000 refer to? If I am right the lower panel is no more than another zoom in on a rectangular area in the top panel. Better avoid the scale terminology. From the bottom panel it also seems that there is some spatial variation in predictions within polygons, although I am not certain because I do not know where polygon boundaries are. If this is true, what is the cause of this (assuming covariates were constant within polygons)?

Reviewer 3 ·

Basic reporting

Digital mapping of soil texture in ecoforest polygons in Quebec, Canada

Dear Editor
I have read above article very carefully. I have the following major comments.

Abstract
Which models?
Please add variables in this part.
Any keywords?
Introduction
Please order citations from old to new.
Literature review needs to some new citations during 2019-2021.
Introduction is written too long.
Please explain about innovative of this study.

Study area
Please add coordinate system of the study area.
Material and methods
Please add a flowchart for the current study.
Covariates
It is important to explain about the applied variables and explain about preparing these factors.
What is its scale or spatial resolution?
Any multi-collinearity test on the applied variables?
Results
This part is written so poor and weak.
Discussion
It is important to add some comparisons with previous publications, advantages, disadvantages, suggestions and future tasks.

Experimental design

please consider my comments.

Validity of the findings

please consider my comments.

Additional comments

Digital mapping of soil texture in ecoforest polygons in Quebec, Canada

Dear Editor
I have read above article very carefully. I have the following major comments.

Abstract
Which models?
Please add variables in this part.
Any keywords?
Introduction
Please order citations from old to new.
Literature review needs to some new citations during 2019-2021.
Introduction is written too long.
Please explain about innovative of this study.

Study area
Please add coordinate system of the study area.
Material and methods
Please add a flowchart for the current study.
Covariates
It is important to explain about the applied variables and explain about preparing these factors.
What is its scale or spatial resolution?
Any multi-collinearity test on the applied variables?
Results
This part is written so poor and weak.
Discussion
It is important to add some comparisons with previous publications, advantages, disadvantages, suggestions and future tasks.

---

## Round 0.2 · Minor Revisions

The authors have made a thorough and mostly satisfactory revision of the manuscript. However, there are still some concerns of two reviewers regarding the manuscript. So I recommend minor revisions and ask the authors for an extra effort to address the reviewers' concerns.

Besides the reviewers' comments, I have very few minor comments:

- In line 199, use capital letters for the institution: Direction de la Recherche Forestière.
- In Table 1, "mesic" is not moisture but temperature regime.

Reviewer 1 ·

Basic reporting

see general comments

Experimental design

see general comments

Validity of the findings

see general comments

Additional comments

The authors have addressed the concerns of the reviewers adequately with one exception which follows:

The authors don’t appear to have carried out any form of external validation of their model. They carried out internal validation using 5 repeats of 5-fold cross-validation, but this was used to tune parameter values. They should have used a test data set that was not used in the internal validation. Ideally, the authors should validate their model with an independent dataset, however, since this is likely not feasible at this point, they should discuss this limitation of their study in their discussion section.

Reviewer 2 ·

Basic reporting

See below.

Experimental design

See below.

Validity of the findings

See below.

Additional comments

I am satisfied with most of the authors’ responses to my comments to the original manuscript and how they incorporated these in the revision. I do not agree with everything they write, but in many cases this is a matter of opinion and I accept that different views exist. I did get the impression that authors are sometimes too easily satisfied and could have done more with my suggestions.

There are some points that I do need to mention explicitly and that in my view may call for a new revision round:

1. I still have serious concerns about how authors dealt with the vertical dimension (depth) in this study. For example, Figure 7 shows maps of the soil texture composition of the diagnostic B horizon but large parts of the study area do not have a diagnostic B horizon. So how to interpret this map? How can one predict soil texture of a horizon that is not there? The escape route that authors use in their response is that in fact soil texture does not vary much with depth so it really does not matter that much at what depth soil texture is predicted, and whether it is the B or C horizon or any other horizon. I find this unsatisfactory, much too easy, and is it really true? If that were the case why not forget about depth altogether, ignore horizons and simply predict in 2D and use calibration data from all depths? Can authors provide some evidence that shows that vertical variation in soil texture is negligible (note that they write themselves that clay can accumulate)? Please note that I am not asking that authors do a 3D modelling exercise because I fully agree that having observations at only two depths per sampling location is not enough for calibration of a 3D model. But there are some important methodological problems with predicting soil texture for the diagnostic B and C horizons, as done in this paper.

2. “For some unknown reason, 33 sites have 2 samples in the database” (L194). Is this not very worrying? Should you not check why this is the case? After all, you selected observations from the first diagnostic B soil horizon only, so there can be only one? Or were duplicates taken, to quantify micro-scale variation and/or lab errors? I would want to know what is the cause of 33 sites having 2 samples in the database because it may also point to serious errors in the database (wrong attribution of soil samples to sampling locations). Why are authors so easily satisfied, why don’t they go to the bottom of this?

3. In their response authors make clear that the soil characteristic covariates were measured at sample sites and characterized by photointerpreters at the prediction locations. This is worrying too, because it effectively means that different covariates were used for calibration than for prediction. For prediction, we have only expert judgements of the ‘true’ covariates, and experts make errors (and must ignore spatial variation within polygons). All this affects the results, in particular the uncertainties (prediction intervals). These will be too ‘optimistic’ because the model assumes that the ‘true’ covariates are known at prediction sites while all that we have are proxies of them. For a paper that addressed this problem in a case where the covariate was soil type (as measured on sampling sites, as depicted on a soil type map at prediction sites), see model 3 in Kempen et al. (2010), European Journal of Soil Science 61, 333-347.

4. Authors confirm that the R-square is defined as the square of the Pearson correlation. I find this very unfortunate, because this does not convey how close the predictions are to the observations. For instance, in all cases where prediction=a+b*observation we will get an R-square of 1, regardless the value of a and b (except when b=0). So even if there is a strong bias (intercept a different from zero) and smoothing (slope b smaller than 1) we still would conclude that the model performance is perfect. Clearly undesirable. Authors defend their choice because this is how it is done in the postResample function of the caret R package, but do they seriously believe that this is a valid justification? If others do things wrongly then they can do it as well? Again I think authors are much too easily satisfied. Instead they should help correcting this mistake and clarify to the DSM research community that in case of (cross-)validation, where we compare predictions with observations, performance should be evaluated against the 1:1 line instead of against the fitted linear regression between predicted and observed. For this we can use Lin’s concordance correlation or the Model Efficiency Coefficient, as I had suggested in my review of the original manuscript. Perhaps in this study the differences may not be very large, but in other cases it will and we should not misguide the readers.

5. Authors prefer to keep the reference to cartographic scale in Figure 8 because “this is a common element of a map”. They explain it now and write in the caption that scale 1:200,000 means that 1cm on the map represents 2km on the ground. So when the article is published, should I measure the distance in cm on the computer screen to derive the distance on the ground? What if I zoom in on the PDF? What if I use a different computer with a bigger screen? I am sorry for being cynical, but if authors had given this a bit more thought they would have realised that cartographic scale only works well with printed maps.

Reviewer 3 ·

Basic reporting

Dear Authors
Thank you so much for revised version. I have read considered the revised version. I am positive with this version. My decision is accept in the current form.

Experimental design

Experimental desinged very-well.

Validity of the findings

good.

Additional comments

Thanks for revised version.

---

## Round 0.3 · accepted · Accept

I am pleased to see that the reviewers are satisfied with the revision of the manuscript and to communicate you that the article is now accepted for publication in PeerJ.

Reviewer 1 ·

Basic reporting

Fine

Experimental design

Fine

Validity of the findings

Fine

Additional comments

I suggest the paper now be accepted.

Reviewer 2 ·

Basic reporting

See below.

Experimental design

See below.

Validity of the findings

See below.

Additional comments

I am happy with this second revision and response to my comments and have no further comments.